# The effect of soil properties on zinc lability and solubility in soils of Ethiopia—

# an isotopic dilution study

Mossa AW[a], Gashu D[b], Broadley MR[a], Dunham SJ[c], McGrath SP[c], Bailey EH[a]*, Young SD[a]

[a]School of Biosciences, University of Nottingham, Sutton Bonington Campus, Nottinghamshire LE12 5RD, UK

[b]Centre for Food Science and Nutrition, Addis Ababa University, P.O. Box 1176, Addis Ababa, Ethiopia

[c]Sustainable Agriculture Sciences Department, Rothamsted Research, Harpenden, Hertfordshire, AL5 2JQ, UK

*Correspondence: EH Bailey, e-mail: liz.bailey@nottingham.ac.uk

**ABSTRACT**

Zinc (Zn) deficiency is a widespread nutritional problem in human populations, especially in sub-Saharan Africa (SSA). The Zn concentration of crops consumed depends in part on the Zn status of soil. Improved understanding of factors controlling the phyto-availability of Zn in soils can contribute to potential agronomic interventions to tackle Zn deficiency, but many soil types in SSA are poorly studied.

Soil samples (n = 475) were collected from a large part of the Amhara Region of Ethiopia where there is widespread Zn deficiency.  Zinc status was quantified by measuring several fractions: pseudo-total (Aqua-Regia digestion; $Zn_{Tot}$), available (DTPA-extractable; $Zn_{DTPA}$), soluble (dissolved in 0.01 M $Ca(NO_3)$; $Zn_{Soln}$) and isotopically exchangeable Zn using the enriched stable Zn isotope $^{70}Zn$ ($Zn_E$). Soil geochemical properties were assessed for their influence on Zn lability and solubility. A parametrised geochemical assemblage model (WHAM7) was also employed to predict the solid phase fractionation of Zn in the soils under study, as an alternative to sequential chemical extractions.

$Zn_{Tot}$ ranged from 14.1 to 291 mg kg$^{-1}$ (median = 100 mg kg$^{-1}$) whereas $Zn_{DTPA}$ in the majority of soil samples was less than 0.5 mg kg$^{-1}$ indicating widespread phytoavailable Zn deficiency in these soils. The labile fraction of Zn in soil ($Zn_E$ as %$Zn_{Tot}$) was low, with median and mean values of 4.7% and 8.0 % respectively. Labile Zn partitioning between the solid and the solution phases of soil was highly pH-dependent where 94% of the variation in the partitioning coefficient of $^{70}Zn$ was explained by soil pH. Similarly, 86% of the variation in $Zn_{Soln}$ was explained by soil pH.

Zinc distribution between adsorbed $Zn_E$ and $Zn_{Soln}$ was controlled by pH. Notably, Zn isotopic
exchangeability increased with soil pH. This contrasts with literature on contaminated and urban soils
and may arise from covarying factors such as contrasting soil clay mineralogy across the pH range of
the soils used in the current study. These results could be used to improve agronomic interventions
to tackle Zn deficiency in SSA.

## 1. Introduction

Zinc deficiency is a widespread nutritional disorder affecting ~17% of the global population, and rising
to 25% of the population in countries within sub-Saharan Africa (SSA) (Kumssa et al., 2015; Wessells
and Brown, 2012). Several interlinked causes contribute to the prevalence of Zn deficiency issues in
SSA, including lack of access to animal source foods. This can lead to inadequate Zn intake if the diet
is heavily reliant on staple crops which are inherently low in mineral micronutrients (Joy et al., 2014;
Kumssa et al., 2015). Soil degradation and a lack of access to micronutrient fertilizers can contribute
to the production of staple crops with poor nutritional quality (Kihara et al., 2020). Three-quarters of
the arable land in SSA is reported to be depleted in plant nutrients and low in fertility (Toenniessen et
al., 2008). However, trace metal dynamics in SSA soils are rarely studied. For instance, we conducted
a simple search on Web of Science database using the key words "zinc solubility" and "soil" between
2010 and 2021. The search yielded 24 publications, none of which involved SSA soils. This is potentially
a serious omission because Zn geochemistry in SSA soils is likely to differ from that in temperate soils
because of differences in geocolloidal minerology, organic C content and the soil pH at which
agriculture is practiced.

Phyto-availability of Zn in soil is largely controlled by a dynamic equilibrium between the solid phase
and pore water and the absorption mechanisms of plant roots (Groenenberg et al., 2010; Menzies et
al., 2007; Peng et al., 2020). Traditionally, chemical extraction procedures used to estimate an
assumed 'phyto-available' pool of soil Zn have included reagents which vary widely in extraction
power such as water, neutral salt solutions, dilute strong acids and chelating agents such as
ethylenediaminetetraacetic acid (EDTA) and diethylenetriaminepentaacetic acid (DTPA; Kim et al.,
2015). However, these approaches cannot fully characterise both the 'quantity' of potentially available
Zn in the soil solid phase and its 'intensity' in the soil solution phase—both of which contribute to the
phyto-availability of Zn over the course of a growing season. Isotopic dilution assays may provide a
more mechanistically-based characterization of the geochemically reactive fraction of Zn in soils which
buffers the free ion activity in the soil solution phase (Guzman-Rangel et al., 2020; Hamon et al., 2008;
Young et al., 2005). This approach has been extensively used to study contaminated soils (Degryse et

al., 2011; Izquierdo et al., 2013; Mossa et al., 2020; Nolan et al., 2005) but its application to Zn in agricultural soils generally, and especially in the soils of SSA countries, is very limited.

The aim of the present study was to investigate the status of Zn lability in soils from a large part of the Amhara Region of Ethiopia which represent a diverse range of soil types from SSA (Gashu et al., 2020), in which Zn deficiency is thought to be widespread (Hengl et al., 2017). The study used several assays of soil Zn status including an isotopic dilution assay, employing enriched $^{70}$Zn, to examine the soil properties that control Zn phyto-availability. The primary objectives were: (i) to determine isotopically exchangeable Zn in soils from the Amhara Region; (ii) to compare different assays of Zn status; (iii) to examine the influence of soil properties on Zn partitioning between the solid and solution phases of these soils; (iv) to investigate the best of three possible input parameters defining reactive Zn in tropical soils (isotopically exchangeable, DTPA-extractable, total) when using a parameterised geochemical assemblage model to predict Zn solubility—assumed likely to be the principal driver for plant uptake.

## 2. Material and methods

### 2.1. Soil sampling

Field sampling is described in detail in Gashu et al., (2020). Briefly, topsoil was collected from a target of 475 locations in the Amhara Region of Ethiopia (Fig. 1) according to a geospatial design intended to explore spatial variation in soil and crop properties. The sample frame was constrained to sites where the probability that the land was in agricultural use was ≥0.9. At each sampling location, five sub-samples of topsoil were collected from a 100 m$^2$ circular plot using a Dutch auger with a flight length of 150 mm and a diameter of 50 mm. Any plant material was removed and the five sub-samples were combined, oven-dried at 40 °C for 24–48 hours depending on the moisture content of the soil samples, sieved to <2 mm and homogenised prior to analysis.

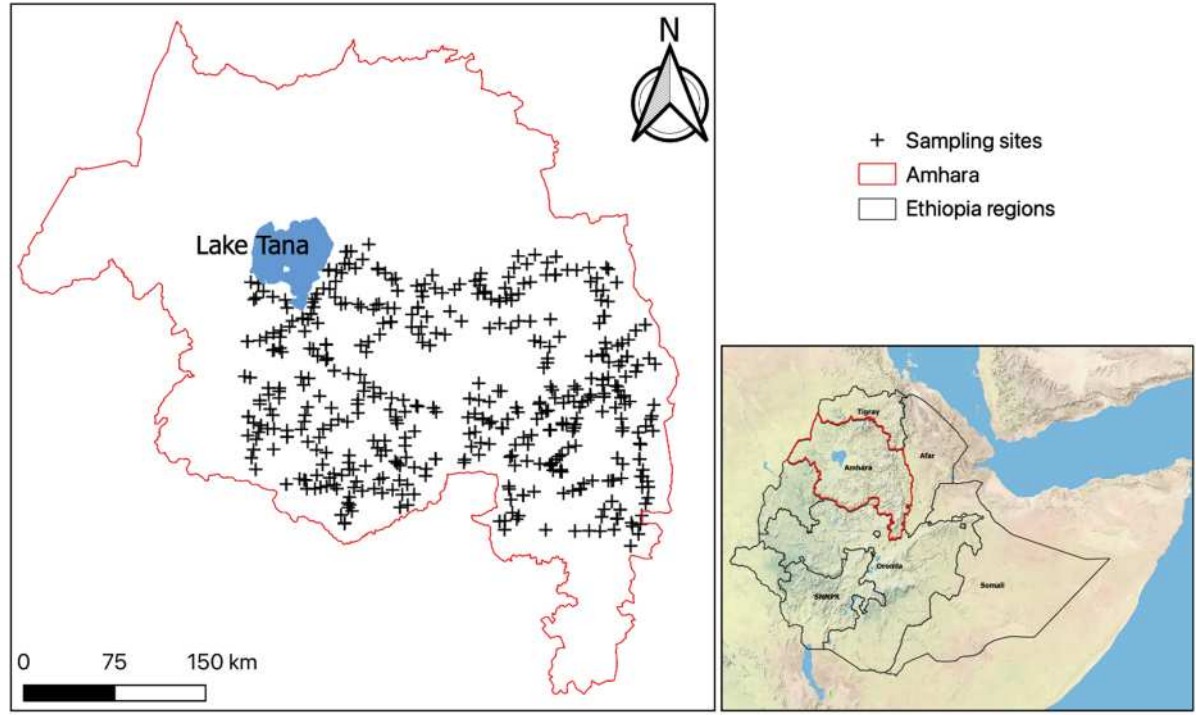

Figure 1. Location of sample sites (black crosses) relative to the border of Amhara region (red line), Ethiopia

## 2.2. Geochemical analysis

Soluble trace and major metallic elements ($M_{Soln}$) were determined in the solution phase of soil suspensions in 0.01 M $Ca(NO_3)_2$ (1:10 soil:solution ratio) following equilibration for four days on an
end-over-end shaker. The pH of this soil suspension ($pH_{Ca}$) was determined, then solutions were isolated by centrifugation and filtration (<0.22 µm) prior to elemental analysis by ICP-MS (iCapQ; Thermo Fisher Scientific, Bremen, Germany). The main parameters of ICP-MS are presented in Table S1. Total carbon content was determined by dry combustion (Tiessen et al., 1981) using a Leco TruMac CN Combustion analyser and inorganic C was measured using an Inorganic Carbon Analyser- Skalar
Primacs (Skalar Analytical BV, Breda, Netherlands). Dissolved organic carbon (DOC) was estimated by measuring non-purgeable organic carbon using a Shimadzu TOC-VWP analyzer (Shimadzu Corporation, Kyoto, Japan). Estimates of amorphous and poorly crystalline oxides were obtained following extraction with a mixture of ammonium oxalate and oxalic acid at a 1:100 of soil:solution suspension (Schwertmann, 1964). Samples were shaken in the dark at 20 °C for 4 hours on a reciprocal
shaker (120 rpm), then filtered (Whatman No 42), diluted and acidified to 5% $HNO_3$, and analysed using inductively coupled plasma optical emission spectrometry (ICP-OES; Perkin Elmer Life and Analytical, Shelton, USA). The effective cation exchange capacity (eCEC) was determined using the cobalt hexamine (Cohex) method (ISO 23470; 2018). DTPA-extractable zinc ($Zn_{DTPA}$) was solubilized by shaking 5 g soil with 10 mL of 0.005 M DTPA, 0.1 M triethanolamine (TEA) and 0.01 M $CaCl_2$ at pH =

7.3 for 2 h on an end-over-end shaker (Lindsay and Norvell, 1978). The soil suspensions were then centrifuged and filtered (<0.22 µm) prior to analysis using ICP-MS (iCAP-Q; Thermo Fisher Scientific, Bremen, Germany). The pseudo-total Zn concentration in soil ($Zn_{Tot}$) was determined after digesting finely ground soil sample with aqua-regia McGrath and Cunliffe, 1985) and analysis using ICP-OES. Blanks and in-house standards were included in each extraction run and calibration standards were

checked using independent certified calibration verification standard solutions. Soil reference material ISE 962 purchased from Wageningen Evaluating Programs for Analytical Laboratories (WEPAL) was used to confirm the reliability of the aqua regia extractions. Recovery of Zn was 91.3% ± 2.35%. Repeat extractions and analysis was undertaken on 10% of the samples.

### 2.3. Isotopic dilution assays

To determine the concentration of isotopically exchangeable Zn, 2.0 g of soil was equilibrated with 20 mL of 0.01 M $Ca(NO_3)_2$ on an end-over-end shaker for 24 h. This was followed by addition of 0.4 mL solution with a $^{70}Zn$ concentration of 11.9 mg $L^{-1}$ of $^{70}Zn$ which provided an amount of $^{70}Zn$ equivalent to 2.3% of the average $Zn_{Tot}$ (104 mg $kg^{-1}$) concentration in soil. The isotopic tracer solution was prepared from a stock solution enriched with $^{70}Zn$ (250 mg $L^{-1}$; isotopic abundance (IA) = 95.47%). To

avoid acidifying the soil suspension, the pH of the spiking solution was adjusted to pH 4.0–4.5 using an ammonium acetate buffer immediately before use. After spiking with the isotope, samples were re-equilibrated for a further 3 days, then centrifuged (3500 rpm for 15 min), filtered (< 0.22 µm), and acidified to 2% $HNO_3$ prior to isotopic analysis using ICP-MS (iCAP-Q; Thermo Fisher Scientific, Bremen, Germany). The instrument was operated in 'collision cell mode' using He with kinetic energy

discrimination (KED). Rhodium ($^{103}Rh$; 10 µg $L^{-1}$) was used as an internal standard to correct for instrumental drift. The ICP-MS was calibrated for individual Zn isotopes ($^{66}Zn$ and $^{70}Zn$) using a multi-isotope Zn standard (CLMS2; SPEX CertiPrep). In practice, it was found that the measurement of $^{70}Zn$ in the supernatant solution of the soil suspensions required two corrections due to significant, and variable, concentrations of soil-derived $^{70}Ge^+$ and (plasma-generated) doubly-charged $^{140}Ce^{++}$ (m/z =

70). The correction for $^{70}Ge$ (IA = 69.9%) was achieved by determining the intensity (count per second (CPS)) of $^{72}Ge$ in samples and using the measured CPS ratio 72/70 for Ge standards to infer the proportion of the intensity at m/z 70 arising from $^{70}Ge$. The universal isotopic ratio $^{72}Ge/^{70}Ge$ is 1.34; the measured intensity ratio in a Ge standard (including error due to mass discrimination) was typically 1.53. The correction for doubly-charged $^{140}Ce^{++}$ was implemented by running Ce standards, which

typically produced a 70/140 intensity ratio of 0.025, and measuring m/z 140 on samples. The Ce standards were analysed in three concentrations of NaCl (0, 1, and 10 mg $L^{-1}$) to confirm minimal change in the generation of doubly-charged Ce in the plasma with alkali cation concentration. The correction for $^{70}Ge$ produced a change in $Zn_E$ that ranged from 0.027 mg $kg^{-1}$ to 0.976 mg $kg^{-1}$ (median

= 0.253 mg kg$^{-1}$) while for $^{140}Ce^{++}$ the change was 0.024 mg kg$^{-1}$ to 0.973 mg kg$^{-1}$ (median = 0.747 mg kg$^{-1}$).

The E-value of Zn ($Zn_E$; mg kg$^{-1}$) was calculated from Eq. 1:

$$Zn_E = Zn_{Soln}\left(Kd_{Lab} + \frac{L}{S}\right) \qquad (1)$$

Where $Zn_{Soln}$ is the measured concentration of Zn of an equilibrated soil suspension; $\frac{L}{S}$ the liquid to solid ratio (L kg$^{-1}$); $Kd_{Lab}$ is the distribution coefficient (L kg$^{-1}$) of the added $^{70}Zn$ isotope spike between a weight of soil (S; kg) and volume of liquid (L) and is calculated as shown in Eq. 2.

$$Kd_{Lab} = \frac{^{70}Zn_{Ads}}{^{70}Zn_{Soln}} \qquad (2)$$

The variable $^{70}Zn_{ads}$ is the adsorbed $^{70}Zn$ spike (µmol kg$^{-1}$) and was calculated as the difference between the total $^{70}Zn$ added to the soil suspension and the amount of $^{70}Zn$ spike remaining in the solution after equilibration; $^{70}Zn_{Soln}$ is the measured concentration (µmol L$^{-1}$) of $^{70}Zn$ spike in solution after equilibration. Crucially, the value of $^{70}Zn_{Soln}$ was corrected for the presence of native $^{70}Zn$ in solution which was estimated from the measured concentration of $^{66}Zn$ and the known isotopic ratio $^{70}Zn$:$^{66}Zn$; this was implemented after all analytical corrections ($^{70}Ge$ and $^{140}Ce^{++}$) and calculation of the Zn isotope concentrations (µmol L$^{-1}$) from isotopic calibration. The measured $^{70}Zn$ was overwhelmingly (97.7% ± 2.20%) dominated by the added spike. Therefore, any deviation from the expected isotopic ratio of $^{70}Zn$:$^{66}Zn$ in the native soil Zn of individual samples would incur a negligible error.

### 2.4. Geochemical modelling using WHAM7

The geochemical model WHAM7 (Tipping, 1994) was used to speciate Zn in the solution phase of the 0.01 M Ca(NO$_3$)$_2$ suspensions. Inputs to the model included solution concentrations of cations (Na, Mg, Al, K, Ca, Mn, Fe, Co, Ni, Cu, Zn, Cd, Ba, Pb, U) and the anions (NO$_3^-$, PO$_4^{-3}$) in the solution phase of the Ca(NO$_3$)$_2$ soil suspensions, colloidal (dissolved) fulvic acid, pH, temperature and partial pressure of CO$_2$. The oxide phases Al(OH)$_3$ and Fe(OH)$_3$ were allowed to precipitate. Colloidal fulvic acid (FA) was estimated from non-purgeable organic carbon (NPOC) assuming (i) a carbon content in FA of 50% and (ii) that 65% was 'active' (Lofts et al., 2008). Partial pressure of CO$_2$ (PCO$_2$) and the temperature were set to 0.004 atm and 25°C respectively. WHAM7 was also used to predict the labile pool of Zn ($Zn_E$) within the soil particulate phases. This required inclusion of suspended particulates, calculated from 2 g solid in 20 mL of electrolyte, and included Fe, Al and Mn oxides (estimated by oxalate extraction) and particulate humic acid which was estimated from soil organic C assuming 50% is 'active' humic acid. The surface chemistry of oxides is simulated by a surface complexation model

(Lofts and Tipping, 1998), which views the oxide surfaces as bearing hydroxyl groups that interact with protons and metal ions. Default values for specific surface areas in WHAM7 were used.

## 2.5. Data analysis

Data analysis was carried out using R (version 4.0.2) software (R Core Team, 2020). Measured soil properties were related to Zn lability ($Zn_E$) and the labile distribution coefficient of $^{70}Zn$ ($Kd_{Lab}$) using standard least squares regression. Soil variables used in the regression were: soil pH (measured in the $Ca(NO_3)_2$ suspensions), organic C (%), sum of the concentration of Al, Fe and Mn in the oxalate extractions (mol kg$^{-1}$), dissolved organic C (mg L$^{-1}$) and the effective cation exchange capacity (eCEC; cmol$_c$ kg$^{-1}$). All data were checked for normality using the Shapiro-Wilk normality test and log-transformed when necessary.

## 3. Results and Discussion

### 3.1. General soil properties

Fluvisols, Leptosols, Lixisols, Luvisols and Vertisols are the prominent soil types in the study area (Dewitte et al., 2013). Most measured soil properties varied widely (Table 1). Soil pH ranged from 4.2–7.5 with ca. 70% soils having pH values below 6.0. The organic carbon ($C_{Org}$) also varied widely with a median value of 1.72% (Table 1). There was a 10-fold variation in eCEC, potentially indicating a large range of Zn binding strength within the studied soils.

Table 1. Selected properties for soil samples (n = 465).

| | Minimum | Maximum | Median | Mean | Std.dev. |
|---|---|---|---|---|---|
| pH$_{Ca}$ | 4.2 | 7.5 | 5.5 | 5.6 | 0.8 |
| Total N (%) | 0.03 | 0.67 | 0.15 | 0.17 | 0.10 |
| Total C (%) | 0.27 | 8.40 | 1.70 | 1.90 | 1.20 |
| $C_{org}$ (%) | 0.27 | 8.36 | 1.70 | 1.90 | 1.20 |
| eCEC (cmolc kg$^{-1}$) | 4.96 | 55.8 | 33.5 | 32.2 | 12.6 |
| Ca (cmolc kg$^{-1}$) | 3.12 | 42.8 | 21.0 | 20.5 | 9.39 |
| K (cmolc kg$^{-1}$) | 0.05 | 4.03 | 0.49 | 0.61 | 0.48 |
| Mg (cmolc kg$^{-1}$) | 0.91 | 21.20 | 6.97 | 7.35 | 4.06 |
| Na (cmolc kg$^{-1}$) | 0.03 | 0.64 | 0.08 | 0.09 | 0.06 |
| AlOx (mg kg$^{-1}$) | 969 | 18800 | 3600 | 4010 | 1940 |
| FeOx (mg kg$^{-1}$) | 1030 | 3220 | 9500 | 9740 | 4980 |
| MnOx (mg kg$^{-1}$) | 153 | 4690 | 1430 | 1500 | 686 |
| Olsen P (mg kg$^{-1}$) | 1.40 | 178 | 10.6 | 17.6 | 21.6 |
| Zn$_{Tot}$ (mg kg$^{-1}$) | 14.1 | 291 | 100 | 102 | 24.3 |
| Zn$_{DTPA}$ (mg kg$^{-1}$) | 0.01 | 5.25 | 0.69 | 0.89 | 0.73 |
| Zn$_E$ (mg kg$^{-1}$) | 0.44 | 57.7 | 4.82 | 7.93 | 8.66 |

| Zn$_{soln}$ (mg kg$^{-1}$) | 0.007 | 0.789 | 0.032 | 0.106 | 0.141 |
|---|---|---|---|---|---|

To further illustrate the general status of Zn in the soils, histograms of different indices of Zn lability and solubility are presented in Fig. 2. Total concentration of Zn in soil (Zn$_{Tot}$) ranged from 14.1 to 291 mg kg$^{-1}$. The median value was 100 mg kg$^{-1}$ (Fig. 2A and Table 1), which is at the top of the range suggested for uncontaminated soils: 10—100 mg kg$^{-1}$ (Mertens and Smolders, 2013). Values of Zn$_E$ ranged from 0.44 to 57.7 mg kg$^{-1}$ (median: 4.82 mg kg$^{-1}$) (Table 1 and Fig 2B). The labile fraction (Zn$_E$ as % of Zn$_{Tot}$) ranged from 0.75% to 69.7% with median and mean values of 4.66% and 8.00% respectively. These values (%Zn$_E$) are lower than those reported for both contaminated and uncontaminated soils (Degryse et al., 2011; Izquierdo et al., 2013; Marzouk et al., 2013b). The distribution of Zn$_{DTPA}$ concentrations were positively skewed (Fig. 2C) with a variation of 0.01–5.25 mg kg$^{-1}$ (median = 0.69 mg kg$^{-1}$). Only 31.4% of samples had Zn$_{DTPA}$ less than 0.5 mg kg$^{-1}$, indicating that they are potentially Zn deficient (Mertens and Smolders, 2013). Values of Zn$_{DTPA}$ as % of Zn$_{Tot}$ ranged from 0.013% to 3.82% with a median and mean of 0.690% and 0.871% respectively. There was a significant but weak positive correlation ($r = 0.25$) between C$_{Org}$ and %Zn$_{DTPA}$ possibly indicating that Zn bound to soil organic matter is in a form accessible to DTPA extraction.

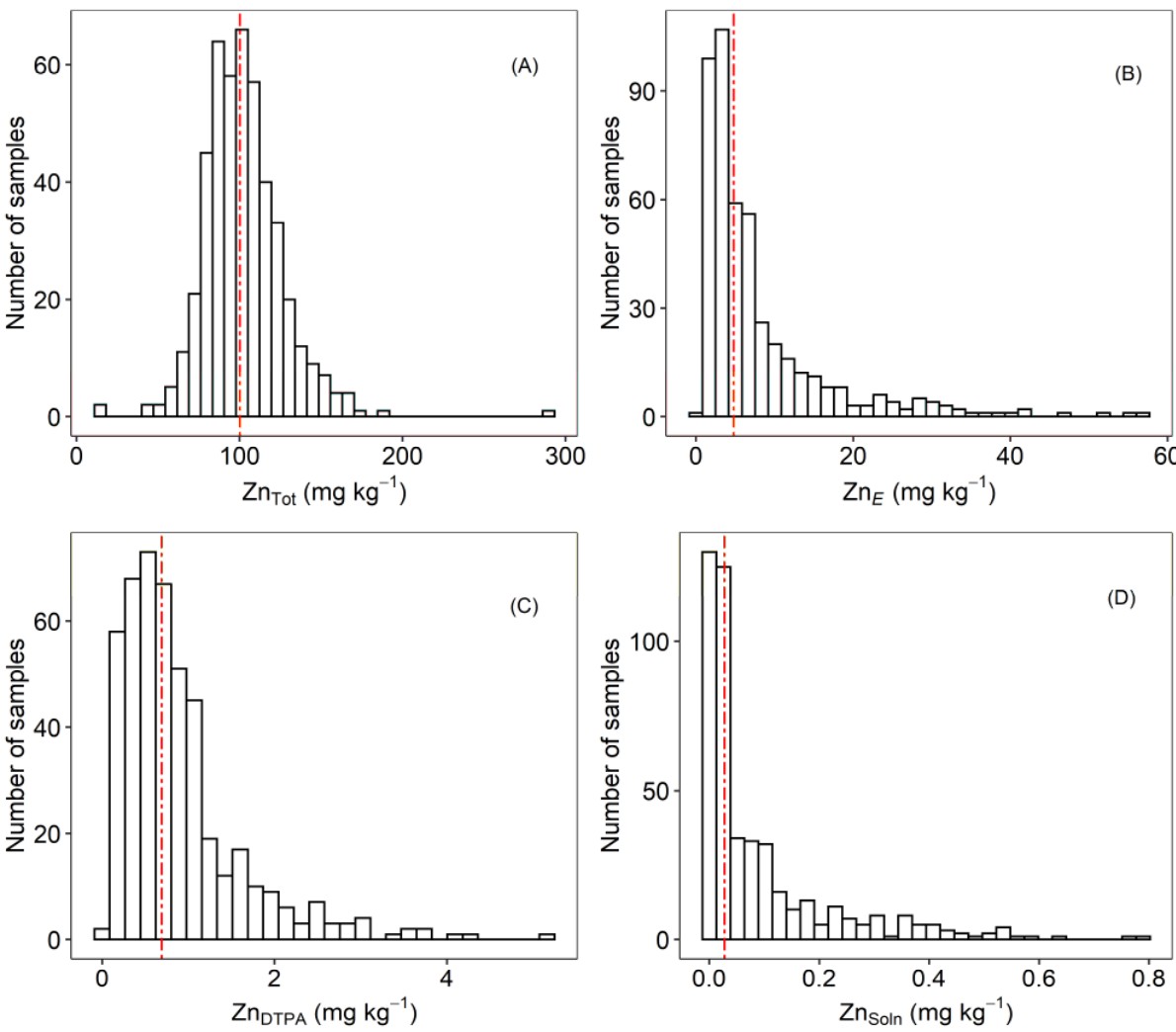

Figure 2. Frequency distributions of (A) $Zn_{Tot}$; (B) $Zn_E$; (C) $Zn_{DTPA}$; (D) $Zn_{Soln}$ concentrations in agricultural topsoil samples from the Amhara region, Ethiopia. Vertical dashed lines represent median values.

The Zn concentration in 0.01 M $Ca(NO_3)_2$, $Zn_{Soln}$, varied by more than two orders of magnitude (0.001–0.789 mg kg$^{-1}$). Values of $Zn_{Soln}$ showed a unimodal and positively skewed distribution (Fig 2D), indicating predominately small concentrations (<0.1 mg kg$^{-1}$ in 72% of soils studied). A maximum of 0.96% of $Zn_{Tot}$ was extracted in $Ca(NO_3)_2$ (median = 0.027%).

    To evaluate the correlation between soil variables, principal component analysis (PCA) and Pearson's

correlation analysis were employed (Fig. 3 and Table S2). The first two principal components (PCA 1 and 2) explained 58.7 % of the variation in the datasets; 41.1% was explained by PCA 1. PCA 1 was strongly correlated with $Kd_{Lab}$ and soil properties that are likely to affect $Kd_{Lab}$, such as pH and eCEC. PCA 2 correlated with $Zn_{DTPA}$, $C_{org}$, and mineral oxides (Fig. 3). PCA analysis also shows that $Zn_{Soln}$ and $Zn_E$ appear to be inversely correlated, which is mainly a consequence of their opposite trends with pH.

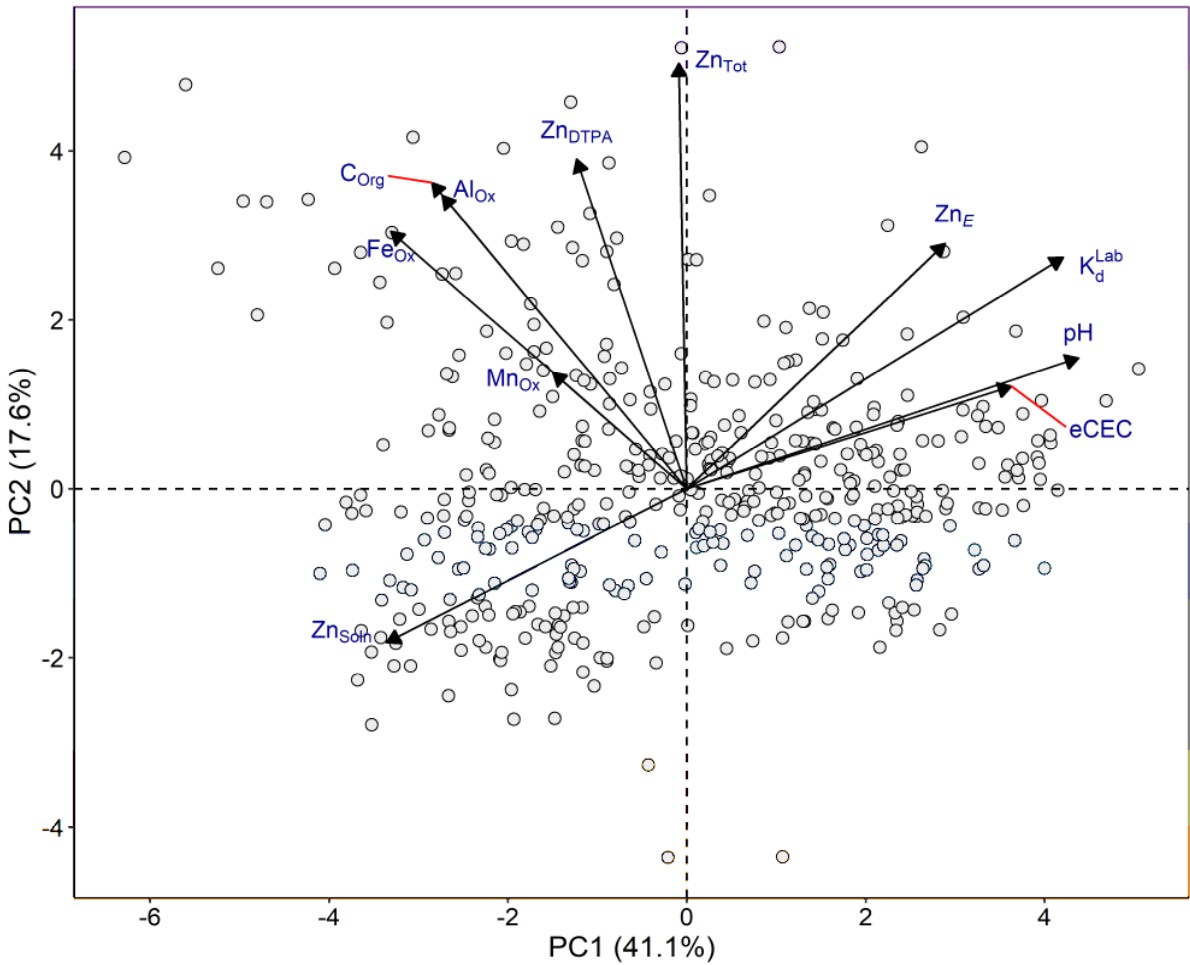

Figure 3. Principal component analysis of main soil variables and Zn indices.

## 3.2. Isotopically exchangeable Zn

### 3.2.1. Method assessment and validation

In principle, *E*-value determination is based on the premise that an added isotope is reversibly adsorbed and is in a dynamic equilibrium between the solid and solution phases (Hamon et al., 2008; Young et al., 2005). Therefore, the reliability of the determined *E*-value rests on an accurate measurement of the distribution coefficient of the added $^{70}Zn$ ($Kd_{Lab}$) and Zn concentration in the soil solution $Zn_{Soln}$ (Eq. 1). For an accurate measurement of $Kd_{Lab}$, the added $^{70}Zn$ must produce a change

in the isotopic ratio ($^{70}Zn/^{66}Zn$) that can be reliably quantified while still reflecting the native Zn equilibrium in the soil. As illustrated in Fig. 4A, there was a clear distinction between the natural isotopic ratio (0.02) and measured ratios, with a minimum $^{70}Zn/^{66}Zn$ ratio of 0.15 which is almost 8 times the natural ratio. At the same time, the amount of the added isotope was small compared to $Zn_{Tot}$ and amounted to 2.1% of $Zn_{Tot}$ on average (< 5% in 94% of the samples). To confirm the

consistency of $Zn_{Soln}$, an inter-laboratory comparison was undertaken. Figure 4B shows the results of

$Zn_{Soln}$ measurements produced by two different laboratories (University of Nottingham and Rothamsted Research) and using different equilibrating electrolytes (0.01 M $Ca(NO_3)_2$ and 0.01 M $CaCl_2$) and different instruments. All data were within one order of magnitude from the line of equality with close agreement ($r$ = 0.96) across the range of the $Zn_{Soln}$ concentrations. Thus, given (i) the robustness of the $^{70}Zn$ distribution coefficient, (ii) the likelihood that the isotopic spike did not substantially affect the native Zn equilibrium or cause precipitation and (iii) the sub-micron filtration step and inter-laboratory agreement for Zn concentration in the 0.01 M Ca electrolyte soil suspensions, we are confident in the validity of the E-value determinations.

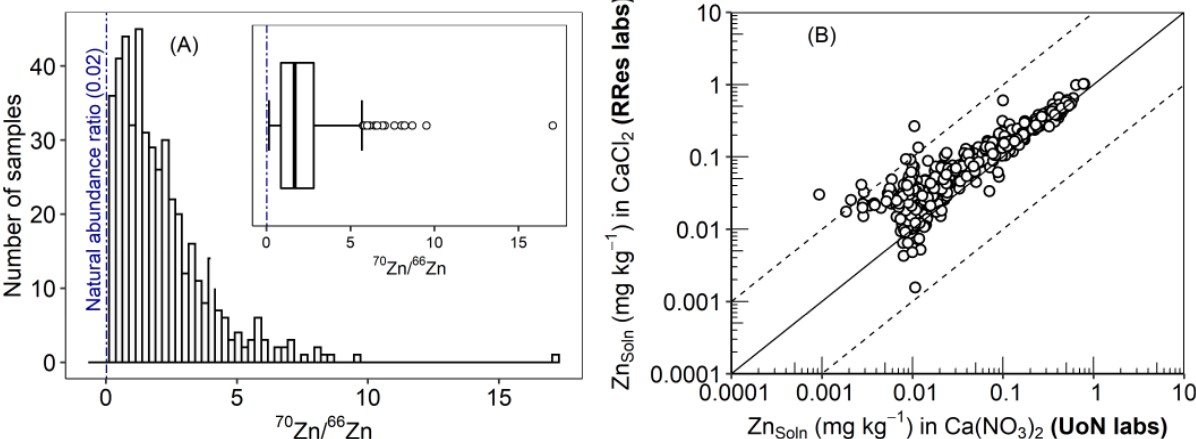

Figure 4. (A) Histogram of the isotopic ratio ($^{70}Zn/^{66}Zn$) in the spiked soils; (B) an inter-laboratory comparison of $Zn_{Soln}$ extracted in 0.01 M $Ca(NO_3)_2$ at the University of Nottingham (x-axis) and in 0.01 $CaCl_2$ at Rothamsted Research (y-axis). The dashed lines and the solid line in B represent 1 log unit interval and the line of equality respectively.

*3.2.2. Soil factors determining Zn lability ($Zn_E$)*

Previous studies, mainly on contaminated soils, have reported that the labile fraction of metals tends to decline with rising pH in response to increased adsorption strength (Degryse et al., 2004; Tye et al., 2003); data in the current study showed the opposite trend (Fig. 5A-B). However, with contaminated soils the behaviour of trace metals often partly reflects the properties of the source of metal (Mao et al., 2014; Marzouk et al., 2013b). For example, contamination with calcareous materials in the case of soils contaminated with mine spoil produces co-variation of total Zn concentration with pH. Furthermore, there is usually a restricted pH range in the case of urban soils and temperate agricultural soils. The current study deals with soils that have comparatively low %$Zn_E$ and $Zn_{Tot}$ and a wide range of pH values (c. 4.0 – 7.5) which are likely to include substantial differences in geocolloidal mineralogy (e.g. oxide-based vs alumina-silicate clays) (Fig. 5C). Thus, the trend depicted in Fig. 5A probably reflects a combination of different factors. For example, in soils with higher pH values it is possible that Zn adsorption is on surfaces which are more likely to retain Zn in a *labile* form—e.g.

humic acid and 2:1 alumino-silicate clays. Similarly, there may be greater Zn fixation under acid conditions because of the greater incidence of oxide-rich mineralogy in highly weathered soils with a low pH (Fig 5C). A significant negative correlation ($r$ = -0.26; $p < 10^{-8}$) between $Zn_E$ and the sum of the concentration of mineral oxides in soil may support that hypothesis. Furthermore, solution phase speciation (from WHAM7) suggested that the proportion of Zn bound to dissolved organic carbon increased with pH (Fig. 6A). At very low Zn concentrations in solution ($Zn_{Soln}$ c. 1 µg L$^{-1}$ above pH 6.5) it is possible that the fulvic-bound Zn was sufficiently strongly bound to be *non-labile* – i.e. excluded from isotopic equilibrium with the added $^{70}$Zn. In the calculation of *E*-value (Eq. 1) this would (erroneously) inflate the apparent $Zn_E$.

A similar outcome would occur if there were significant amounts of non-labile Zn held in particulate form as part of the measurement of $Zn_{Soln}$ at higher pH values. Non-labile particulate metal was first demonstrated by Lombi et al., (2003) who used chelating resin in *E*-values measurements ($E_r$) to quantify the fraction of the colloidal metal that was not isotopically exchangeable. They reported that $E_r$ values were generally less than *E* values based on equilibrated solution measurements (E) and that the ratio $E/E_r$ increased with pH. Use of the resin method has produced variable results. Marzouk et al., (2013b) also reported metals associated with sub-micrometre colloidal particles in the solution phase, based on resin phase measurements. They found this association to be positively correlated with soil humus content and pH. However, for their dataset, they found that the presence of nano-particulate non-labile Zn in solution produced, on average, less than 2% difference in the determination of E-values (E vs $E_r$). Mao et al., (2017) also investigated the presence of non-labile metal fractions of Ni, Cu, Zn, Cd, and Pb in suspended colloidal particles. They also found an average of only 2% difference between $E_r$ and *E* for all five metals and the difference was only significant for Cu with an increased presence of non-labile colloidal particles at high pH – probably organically bound Cu.

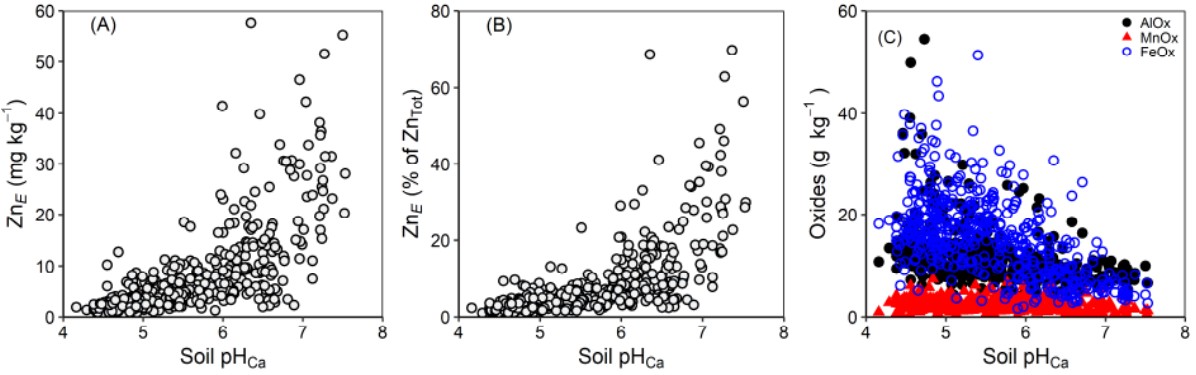

Figure 5. Concentration of (A) $Zn_E$ in soil; (B) $Zn_E$ as % of $Zn_{Tot}$; and (C) mineral oxides in soil, as a function of soil pH$_{Ca}$.

If the trend in $Zn_E$ with soil pH (Fig. 5A-B) was affected by interferences from particulate materials in the soil solution then the source of the error would either be in $Kd_{Lab}$ or $Zn_{Soln}$—the two variables in the calculation of $Zn_E$ (Eq. 1). However, the presence of non-labile Zn within particulate matter in the isolated soil suspension supernatant would not contribute to an error in $Kd_{Lab}$. This is because, by definition, the labile spiked isotope is excluded from mixing with the particulate Zn phase. However, the measurement of $^{66}Zn$ would include Zn in solution and any particulate-bound Zn (< 0.22 μm), with which the $^{70}Zn$ would not have mixed. Thus it is the determination of $Zn_{Soln}$ (Eq. 1) that may produce an error in $Zn_E$. Tavakkoli et al., (2013) investigated the possible occurence of non-isotopically exchangable Zn in sub-micron sized colloids in filtered soil extraction at high soil pH. They found no non-exchangeable Zn when filtering to <0.1 μm to remove particles but gradually increasing proportions of isotopically non-exchangable Zn where solutions had been filtered using progressively larger filter pore sizes (0.22, 0.45, and 0.7 μm). In the present study soil extraction solutions were filtered to <0.22 μm, suggesting that suspended particulates containing non-isotopically exchangeable Zn should be negligible.

The possible presence of non-labile nanoparticulate Zn in the soil solution was investigated using a resin (Chelex-100) purfication step (Marzouk et al., 2013b) in the determination of $Zn_E$. A comparison was made of $^{70}Zn/^{66}Zn$ ratios in the centrifuged, filtered solution and in a resin extraction of that solution. No evidence of non-labile nanoparticulate Zn below pH 5.5 was found; the isotopic ratios $^{70}Zn/^{66}Zn$ in the solution and resin phases were equal. Unfortunately, at higher soil pH (>5.5), our investigation was confounded due to resin Zn contamination that compromised the measurement of low soluble Zn concentrations in soils with high pH. However, considering the strength of the trend depicted in Fig. 5A, the majority of Zn in the filtered soil solution would have to be present as non-labile particulate matter for the trend shown to be due to non-labile particulate Zn contributing to $Zn_{Soln}$. This seems unlikely and so we therefore suggest that the increase in $Zn_E$ values with soil pH in the soils studied is probably a genuine trend.

### 3.3. Zinc solubility

The partition coefficient in the current study ($Kd_{Lab}$) represents the distribution of the added $^{70}Zn$ spike between the isotopically exchangeable Zn on the solid phase and in the solution phase (Eq. 2). Values of $Kd_{lab}$ varied by more than 3 orders of magnitude—ranging from 15.4 to 42600 L kg$^{-1}$. As shown in Fig. 76A, values of $Kd_{Lab}$ were highly pH-dependent, in agreement with increased adsorption strength of cationic trace metals onto soil surfaces with pH. Regression analysis of soil properties (eCEC, $C_{Org}$, $Zn_{Tot}$, mineral oxides) against $Kd_{Lab}$ is presented in Table 2. Only significant variables were retained in regression equations and the variables were checked for multicollinearity using variance inflation factors (VIF). Values of VIF for all variables were less than 3. While all variables in Table 2 were

significant in the regression analysis, they accounted for a very small proportion of the variation (< 4%) in the data. The majority of the variation in the data (90%) was explained solely by soil pH (Table 2 and Fig. 6A). Despite the fact that soil organic matter is known to be an important sorbent for trace elements (Degryse et al., 2011), $C_{Org}$ had a negligible influence on $Kd_{Lab}$.

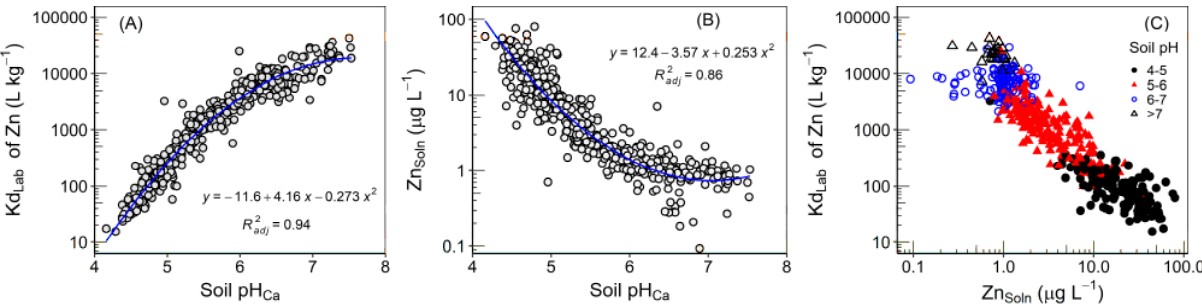

Figure 6. Values of (A) $Kd_{Lab}$ and (B) $Zn_{Soln}$ as a function of soil pH; (C) relationship between $K_d^{lab}$ and $Zn_{Soln}$

As seen for $Kd_{Lab}$, $Zn_{Soln}$ was also mainly controlled by soil pH; 77% of the variation in $Zn_{Soln}$ was explained solely by soil pH (Fig 76B). There was also a weak, but significant, correlation between $Zn_{Soln}$ and soil $C_{Org}$ ($r = 0.23$, $p < 5.8 \times 10^{-7}$); some influence on metal adsorption strength would be expected because of the importance of humus as a metal adsorbent (Fan et al., 2016). However, the limited effect of soil organic matter may be due to a dual influence on Zn solubility. Soil humus will contribute to Zn adsorption within the soil solid phase but also produce greater DOC ($r = 0.47$ between $C_{org}$ and DOC) which will promote dissolved organo-complexation of Zn.

Table 2. Regression equations for $Log_{10}(Kd_{Lab})$

| Regression equation | $R^2_{adj}$ | p |
|---|---|---|
| $Log_{10}(Kd_{Lab}) = 1.0 \times pH_{Ca} + 0.3 \times Zn_{Tot} + 0.3 \times C_{Org} + 0.5 \times eCEC + 0.9 \times oxides - 4.0$ | 0.93 | $< 2.2 \times 10^{-16}$ |
| $Log_{10}(Kd_{Lab}) = 1.0 \times pH_{Ca} - 2.8$ | 0.90 | $< 2.2 \times 10^{-16}$ |

The concentration of Zn in soil solutions is largely determined by the combined influence of soil properties which affect the strength of adsorption, and the total Zn concentration in soil. But, the relationship between $Kd_{Lab}$ and $Zn_{Soln}$ (Fig. 6C) demonstrates the much greater importance of soil characteristics over the influence of $Zn_{Tot}$ in the Amhara soils. In considering the relationship in Fig 6C, it should be emphasised that $Kd_{Lab}$ and $Zn_{Soln}$ are completely independent of each other. The value of $Kd_{Lab}$ is the distribution coefficient of the added $^{70}$Zn isotope and $Zn_{Soln}$ is determined from measured values of $^{66}$Zn; this negates the common, and justified, criticism of such relations in which $Zn_{Soln}$ is the denominator of the Kd which would tend to produce a declining trend with $Zn_{Soln}$. Therefore, the very strong capacity-intensity dependence of the studied soils genuinely reflects control by soil properties over Zn solubility. In particular, for the soils studied, soil pH alone virtually controls the strength of Zn

adsorption and $Zn_{Soln}$ (Fig. 6A&B; Table 2), despite considerable variation in $Zn_{Tot}$ (14.1 – 291 mg kg$^{-1}$; Table 1).

**3.4. Multi-surface modelling of soluble Zn concentration**

It is widely recognised that while the total concentration of an element in soil is important, it is the chemical speciation that plays a key role in determining availability to plants. Despite that, direct measurement of the chemical forms of an element is technically challenging. Therefore, geochemical modelling offers a feasible alternative and has been widely applied to soil (Bonten et al., 2008; Cui and Weng, 2015; Klinkert and Comans, 2020).

The WHAM7 predictions of *labile* Zn distribution among different soil surfaces are presented in Fig. 7B. At low soil pH, the WHAM7 model predicted the sorption to be overwhelmingly onto Mn oxide and humic acids, whereas at intermediate and high pH, humic acid-bound Zn became dominant. (Buekers et al., 2008; Marzouk et al., 2013b). WHAM7 predicts a negligible role for Fe oxide in adsorbing Zn but at pH > 6.5 sorption onto Al oxides was important. The fractionation suggested by the WHAM7 model relates only to labile Zn and does not predict the location of the 'fixed' (non-labile) Zn in the soils.

In common with previous studies, soil reactive organic matter was estimated as 50% of total soil organic C in all cases. However, in reality the composition and the reactivity of soil organic matter will differ between soils and there is currently uncertainty regarding the use of a universal value. Thus, for instance, Van Eynde et al., (2020) found that the average fraction of humic substances in 5 tropical soils was 36%. Consequently, there will be an associated error when relying on such assumptions in the absence of full characterization of soil organic matter in all soils. This is particularly important in the current study because while WHAM has been used and validated in several studies on temperate soils (Buekers et al., 2008; Izquierdo et al., 2013; Mao et al., 2017), to our knowledge, it has not been used on tropical soils. To test the effect of soil organic matter composition on model performance, we ran the model assuming 36% of soil organic C to be 'active' humic substances, as suggested by the data of Van Eynde et al., (2020). The results of these simulations are reported in Fig. S1 in the supplementary material. The change in $Zn_{Soln}$ was pH dependent, ranged from –4.8% to -39.8% of the originally modelled value, and the maximum change occurred at soil c. pH 5.5. These results are relatively small changes considering that $Zn_{Soln}$ covers over 4 orders of magnitude in the entire dataset.

Both amorphous and crystalline Fe oxide are involved in Zn binding in soil. The specific surface area of the crystalline Fe oxide has been assumed to be 1/6 of that of the amorphous Fe oxide (Dijkstra et al., 2004). However, it was previously found that oxalate extractable Fe (amorphous) was smaller than dithionite extractable Fe (crystalline and amorphous) in tropical soils (Van Eynde et al., 2020). For

comparison, model performance was tested using double the amount of amorphous oxides as inputs. The results of these comparisons are presented in Fig. S2 in the supplementary material. When doubling the amount of particulate Fe oxide in WHAM7 inputs, there was on average 3.42%, 5.43%, and 2.56% decrease in predicted $Zn_{Soln}$ when using $Zn_E$, $Zn_{DTPA}$, and $Zn_{Tot}$ as inputs respectively. However, the change was highly pH-dependent, and although the effect was negligible at soil pH < 5.5 (Fig. S2) there were much greater changes in the pH range 6 – 7 up to just over 10% for $Zn_{DTPA}$ model inputs.

In the current study, Mn oxides appear to be an important binding phase for Zn. The binding constant, $pK_{MH}$ for Zn onto Mn oxide in WHAM7 is 1.6. However, Lofts and Tipping, (1998) found that measured binding constants for Zn adsorption by Mn oxide ranged from 0.6 – 4.2. They attributed the difficulty in modelling adsorption onto Mn oxides to sparse adsorption data, uncertainty in Mn oxide properties, and variation among results from experimental data. Therefore, the results of modelling Zn adsorption onto Mn oxides, as predicted by WHAM7, should be viewed with caution.

The speciation of Zn in the soil solution, as calculated by WHAM7, is presented in Fig. 7A. It was predicted that the free Zn ion activity ($Zn^{2+}$) constituted 36.1% to 99.2% (median = 77.1%) of the total $Zn_{Soln}$ and was highly correlated with pH. At pH < 5.5, the majority of $Zn_{Soln}$ was present as the free $Zn^{2+}$ ion (> 61%). This percentage decreased to an average of 49% at soil pH > 7. Previous studies have also shown minimal complexation of Zn in the soil solution at pH < 6.5 (Catlett et al., 2002; Rutkowska et al., 2015). The proportion of the total Zn present as dissolved fulvic acid complexes ranged from 0.81% to 61.2% (median 22.7%). The proportion of FA-complexed Zn increased with increasing soil pH (Fig. 6A); at pH > 7 an average of 47.7% of $Zn_{Soln}$ was apparently complexed to FA. The only inorganic Zn complexes were carbonates and these accounted for < 4% at soil pH > 7.

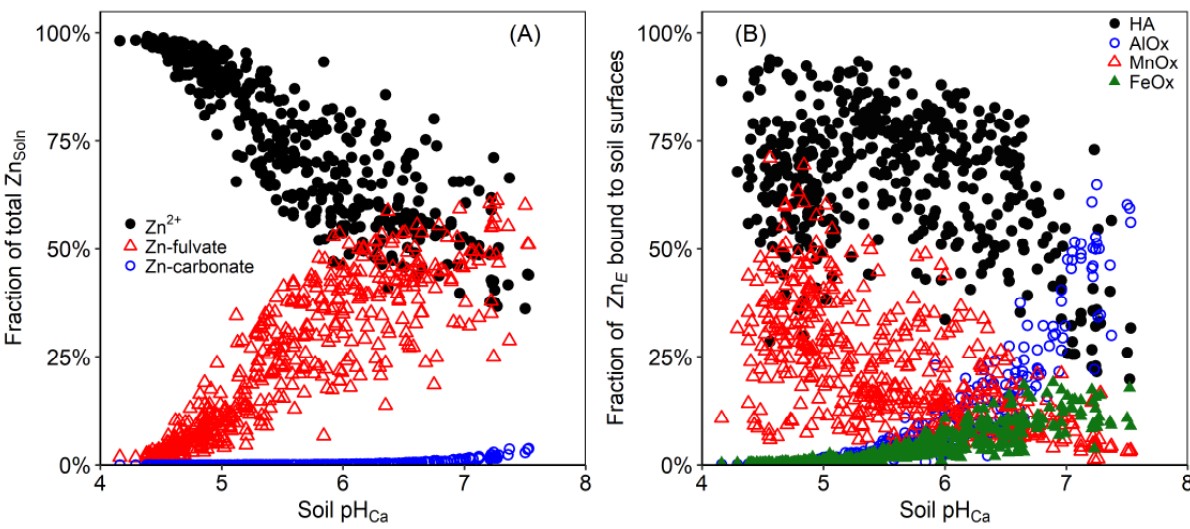

Figure 7. (A) Zn speciation in the *solution* phase of Ethiopian soil suspensions as predicted by WHAM7 (free divalent ions = black circles; FA complexes = red triangles; carbonate complexes = blue circles). (B) Zn fractionation in the *solid phase* as predicted by WHAM7 (HA complexes = black circles; Al oxide bound = blue circles; Mn oxide bound = red triangles; Fe oxide bound = green triangles).

An important consideration when using geochemical models is the choice of the 'reactive' pool of metals, which is in equilibrium with the soil solution, as an input variable. It has been well established that the total concentration of metals does not reflect the reactive fraction in soil (Kelepertzis and Argyraki, 2015; Peng et al., 2018; Rodrigues et al., 2013). Extractions with 0.43 M nitric acid ($HNO_3$) and EDTA have been frequently used to approximate the geochemically reactive pool of metals in soil (Agrelli et al., 2020; Groenenberg et al., 2017; Liu et al., 2019; Ren et al., 2017). However, the isotopic dilution method is recognized to be conceptually the most robust and mechanistically based method that reliably quantifies the reactive pool of metals in soil (Groenenberg et al., 2017; Hamon et al., 2008; Peng et al., 2018). To assess the capability of WHAM7 to predict Zn solubility, the concentrations in the solution phase ($Zn_{Soln}$) were compared with the outputs from fractionation of Zn across the whole soil-solution system, using either $Zn_{Tot}$, $Zn_E$, or $Zn_{DTPA}$ concentrations as the fraction of Zn controlling Zn solubility. Results of these simulations are presented in Fig. 8.

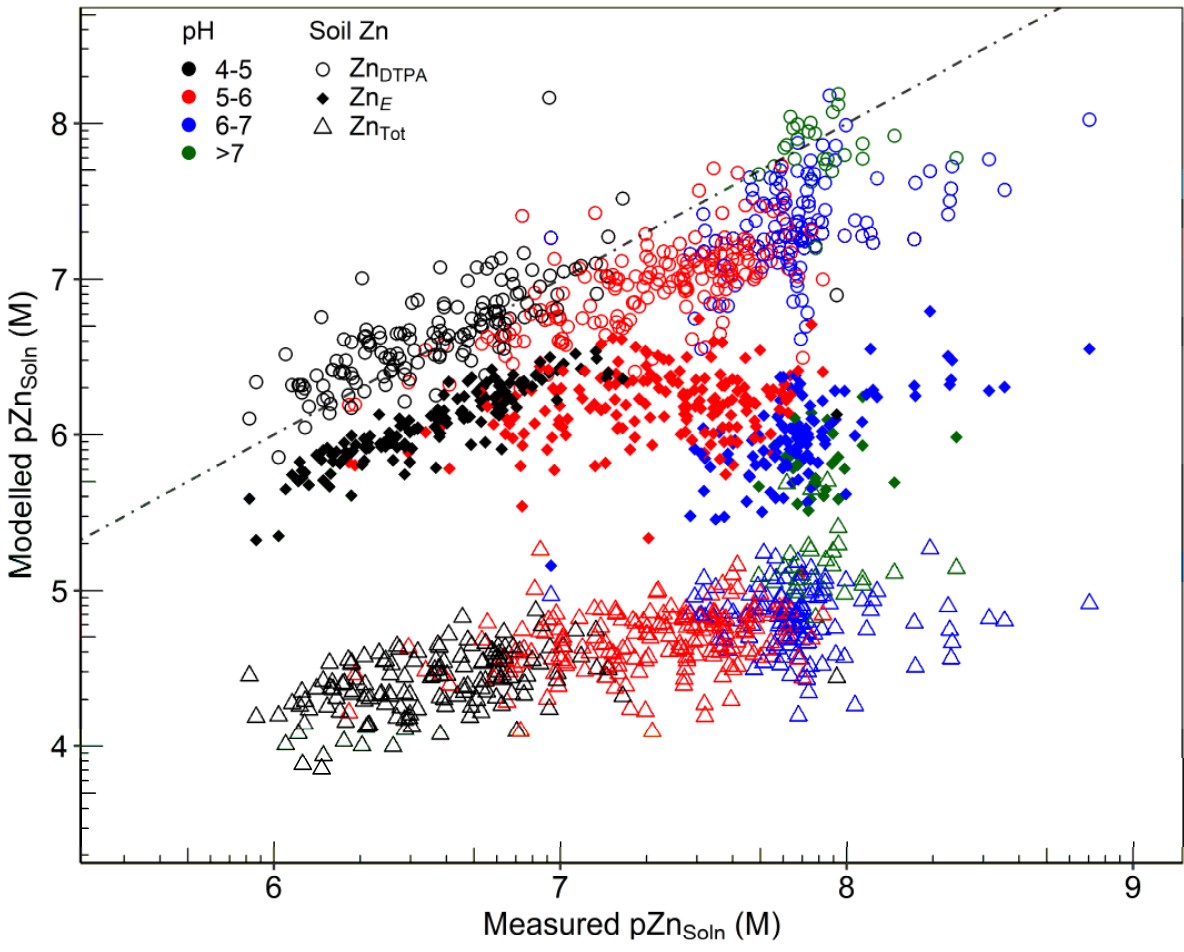


Figure 8. Comparison between observed pZn$_{Soln}$ (X-axis) and modelled pZn$_{Soln}$ (Y-axis) using either Zn$_{DTPA}$, Zn$_E$ or Zn$_{Tot}$ as WHAM7 inputs. The dashed line represents the line of equality.

Figure 8 clearly illustrates that using Zn$_E$, substantially improves the prediction of Zn solubility compared to using Zn$_{Tot}$, particularly at low soil pH. This reinforces the conclusion that the

geochemically reactive metal pool, rather than the total soil Zn concentration, is the most relevant representation of Zn availability in soil. At high pH (>7.5), the model predicts higher Zn$_{Soln}$ than observed concentrations. This may be partly due to limitations in binding surfaces considered in WHAM7. At pH >7, adsorption on calcium carbonate or phosphate minerals may occur which is not accounted for in WHAM7. This was reported by Peng et al., (2018) who excluded data at pH >7 from

their results, when using WHAM7 to predict the solid-solution partition and speciation of heavy metals, in response to a lack of consideration of precipitation on carbonates. Izquierdo et al., (2013) listed the failure to include binding to carbonate surfaces as a possible source of error in predicting metal concentration in the soil solution from WHAM7. Mao et al., (2017) also attributed the overestimation of metal concentrations in the soil solution to the exclusion of phases such as calcite

and hydroxyapatite as binding phases in WHAM7. Additionally, overestimation of $E$-values at high pH due to the presence of (non-labile) Zn which has not isotopically mixed with the added $^{70}$Zn spike

would also explain the poorer performance of WHAM7 in predicting Zn solubility at high pH. Bonten et al., (2008) also reported overestimation of Zn concentration calculated by the geochemical model ORCHESTRA. They pointed out that some authors had suggested sorption of Zn in Al layered double

hydroxides or phyllosilicates, in soils at near-neutral pH, as a potential reason for overprediction of Zn solubility.

When $Zn_{DTPA}$ was used as input, prediction of Zn solubility by WHAM7 was apparently improved over that achieved by using $Zn_E$ (Fig. 8), particularly at high pH. This may confirm the possible over-estimation of $Zn_E$, as discussed above. Alternatively, it may reflect counteracting errors between (i)

the inadequacy of DTPA (0.005 M) as an extractant which would decrease modelled $Zn_{Soln}$ and (ii) the underestimation of Zn binding in WHAM7 at high pH which would raise the estimate of $Zn_{Soln}$. It is recognised, for example, that 0.005 M DTPA extracts less Zn from soil than 0.05 M EDTA and also underestimates $Zn_E$ (Marzouk et al. 2013a).  To assess whether the binding capacity of the DTPA used was limited, the mole ratios of cations to DTPA in the extracted solutions (excluding alkali/alkali-earth

cations) were calculated. The average ratio was only 0.17 ± 0.08, suggesting that the DTPA extractant was probably not capacity-limited. These results suggest, broadly, that both $Zn_{DTPA}$ and $Zn_E$ may be reasonable estimates of the 'labile' pool of Zn in soil. DTPA appears to provide a better estimate of $Zn_{Soln}$ using a current geochemical model, especially at high pH. Alternatively, the isotopic dilution method, measured in neutral 0.01 M $Ca(NO_3)_2$, probably better reflects variation with pH in labile Zn

Kd value, and possibly in the true labile pool of Zn in soil, compared with DTPA, which is buffered at pH 7.3.

### 3.5. Free $Zn^{2+}$ activity in soil solution

The free ion activity is considered a key factor controlling plant uptake, although other factors will affect buffering and diffusion rates in the soil (Degryse et al., 2012). The concentration of $Zn^{2+}$ activity

in the soil solution is effectively an integration of soil properties that govern sorption processes. Data presented in Fig 9 show that the activity of $Zn^{2+}$ is highly pH-dependent; 81% of variation in the free $Zn^{2+}$ activity was accounted for solely by soil pH in the Ethiopian Amhara soils. The concentration of free $Zn^{2+}$ activity varied over 3 orders of magnitude. The range of the free $Zn^{2+}$ activity is probably a product of the counteracting effects of $Zn_E$ and $Zn_{Soln}$ variation with pH; $Zn_E$ increases with pH while

$Zn_{Soln}$ falls as pH rises as discussed above (Fig 5B and 6B).

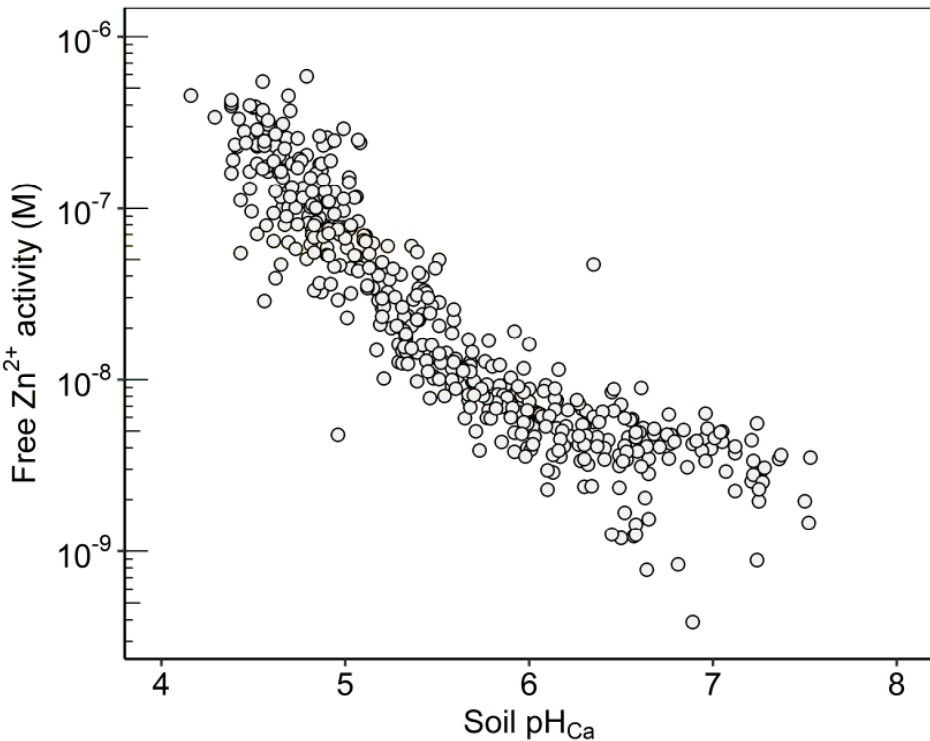

Figure 9. Concentration the free $Zn^{2+}$ activity in the soil solution as a function of soil pH

## 4. Conclusions

In this study combining (i) an isotopic dilution approach to determining reactive soil Zn ($Zn_E$) and the relative strength of available Zn adsorption (Kd) with (ii) geochemical speciation modelling (WHAM7) provided useful insights into the intrinsic reactivity of Zn in SSA soils at a regional scale and revealed the key soil variables determining Zn solubility. In particular, the results demonstrated that intrinsic soil properties, rather than the variation in $Zn_{Tot}$ concentration, were more important in determining the adsorption strength (Kd) of labile Zn and Zn solubility. Specifically, in the Ethiopian Amhara dataset soil pH was the key determining factor. Surprisingly, the traditional DTPA extraction method provided a better estimate of $Zn_{Soln}$, predicted from a geochemical modelling approach, when compared with $Zn_E$ as a model input variable. Reasons for this remain unresolved and may reflect shortcomings in either (i) the crude nature of soil analysis for model inputs or (ii) poorer model prediction at higher pH values due to further soil Zn adsorbents not considered within the WHAM7 model.

These findings may have practical implications for agronomic interventions to improve crop Zn concentrations for they provide a tool for differentiating between soils in terms of both the strength with which they adsorb Zn and their available Zn reservoir. This is an important consideration for a site-specific strategy to ensure more effective agronomic biofortification of staple crops with Zn fertilizers (Joy et al., 2015; Manzeke et al., 2014, 2020; Zia et al., 2020). It may be suggested, for example, that in soils with pH >6.5 foliar Zn application may be most appropriate because Zn is

consistently more strongly adsorbed as pH rises, while in soils with lower pH applying Zn-containing fertilizers to the soil might be feasible. Furthermore, these findings can be used to identify areas where the use of soil management practices, such as liming and organic matter incorporation, are likely to impact Zn availability in soil and thereby affect Zn concentration of staple crops (Manzeke et al., 2019;

Wood et al., 2018).

Zinc uptake by crop plants will be determined by a combination of (i) soil Zn availability and (ii) plant controls over uptake and transport. It is likely that soil Zn availability will reflect both solubility and the magnitude of the reactive Zn reservoir that supports $Zn^{2+}$ ion activity in the soil solution. For the Amhara soils studied it appears that (i) DTPA-extractable Zn provides the best estimate of the available

Zn pool controlling $Zn^{2+}$ solubility and (ii) that pH is the best predictor of the strength of labile Zn adsorption. Both assays are well recognised and fall within the scope of most agronomic laboratories. It therefore seems reasonable to suggest that $Zn_{DTPA}$ and soil pH may, in combination, be tested as predictor variables for Zn uptake by staple crops in considering both crop Zn deficiency and biofortification for human health.

**Data availability**

Data used in this study is available for the corresponding author upon a reasonable request as the lab data will be published as part of a national datasets.

**Author contribution**

Conceptualization of the study for this manuscript was done by AwM, MrB, SpM, and SdY, with input

from EhB, GD, and SjD. Data curation and investigation: AwM. Analysis, methodology, and visualization for the manuscript was performed by AWM with substantial input from MrB, SdY, EhB and feedback from all authors. AWM wrote the initial draft and all authors were involved in the review and editing of the manuscript.

**Declaration of competing interest**

The authors declare that they have no competing financial interests or personal relationships that could have appeared to influence the work reported in this paper.

**Acknowledgements**

This work was supported by 'GeoNutrition' projects, funded by Biotechnology and Biological Sciences Research Council (BBSRC) / Global Challenges Research Fund (GCRF) [BB/P023126/1] and the Bill &

Melinda Gates Foundation (BMGF) [INV-009129]. The funders were not involved in the study design, the collection, management, analysis, and interpretation of data, the writing of the report or the

decision to submit the report for publication. The authors gratefully acknowledge the contribution made to this research by the field sampling team of the Amhara National Regional Bureau of Agriculture; Debebe Hailu and Aregash Beshire also contributed to sample preparation; staff at

Rothamsted who performed many of the extractions and instrumental analyses; and Saul Vazquez Reina at the University of Nottingham who performed the ICP-MS analysis. The authors would like to thank Dr. Diriba Kumssa (University of Nottingham) for providing a map of the sampling area.

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
