# Peer review of "The effect of soil properties on zinc lability and solubility in soils of Ethiopia—"

_SOIL, 2020_

## Author Comment (AC1)

**Responses to Referee 1 : MS No.: soil-2020-81**
Title: Zinc lability and solubility in soils of Ethiopia – an isotopic dilution study
Author(s): Abdul W. Mossa et al.

Many thanks for your time and effort in reading and commenting on our manuscript; your detailed points will help improve it. In the following we address the individual points you raise.

**Referee 1**

1. Introduction (page 1, lines 39-41): this section, describing the quality and fertility of agricultural soils in the sub-Saharan Africa, needs to be widened with a focus on Zn geochemistry.

    **Response:**

    Thank you for your suggestion—we suggest adding a section to highlight the knowledge gap on Zn geochemistry in tropical soils from sub-Saharan Africa compared to temperate soils, as follow:

    > "However, trace metal dynamics in SSA soils are rarely studied. For example, a simple search of the Web of Science database using the key words "zinc solubility" and "soil" between 2010 and 2021 yielded 24 publications, none of which involved SSA soils. This is potentially a serious omission because Zn geochemistry in SSA soils is likely to differ from that in temperate soils because of differences in geocolloidal minerology, organic C content and the soil pH at which agriculture is practiced."

2. Although readers are referred to Gashu et al. (2020) for details on field sampling, I suggest providing a map of sampling area, at least as possible supplementary material. The overall methodology has to be better defined, analysing advantages and possible limitations as well.

    **Response:**

    Thank you for your suggestion— we propose adding a map showing sampling location to the revised manuscript:

[Figure]

Figure 1. Location of sample sites (black crosses) relative to the border of Amhara region (red line), Ethiopia

3. Did authors analyse any certified reference materials? Please provide more details on quality control and quality assurance of soil extractions.

**Response:**

There are no certified materials for most of the extractions used in the study. We used the reference material (ISE 962) provided by Wageningen Evaluating Programs for Analytical Laboratories (WEPAL) for aqua regia extraction. We suggest adding the following details to the revised manuscript:

"Blanks and in-house standards were included in each extraction run and calibration standards were checked using independent certified calibration verification standard solutions. Soil reference material ISE 962 purchased from Wageningen Evaluating Programs for Analytical Laboratories (WEPAL) was used to confirm the reliability of the Aqua regia extractions. Recovery of Zn was 91.3% ± 2.35%. Repeat extractions and analysis was undertaken on 10% of the samples."

4. If possible, gather and show the main operating parameters of the ICP-OES and ICPMS instruments in a table in the supplementary section.

**Response:**

Thank you for your suggestion— we propose to add the following table of main operating parameters of ICP-MS instruments to supplementary material.

**Table S1.** ICP-MS (iCAP-Q) operating conditions for both multi element and isotope ratio

| Parameter | Isotope analysis | Multi-elements analysis |
|---|---|---|
| Dwell time (s) | 0.05 | 0.01 |
| Number of sweeps | 50 | 50 |
| dead time correction (ns) | 34.7 | 34.7 |
| Detector dead time (ns) | 35 | 55 |
| Nebuliser flow rate (L min$^{-1}$) | 1.120 | 1.105 |
| Extraction lens voltage (V) | -176.5 | -111.3 |
| Helium flow rate (L min$^{-1}$) | 4.4 | 4.4 |
| Coolant gas flow (Ar; L min$^{-1}$) | 14 | 14 |
| Auxiliary gas flow rate (L min$^{-1}$) | 0.8 | 0.8 |
| Spray chamber temperature (°C) | 2.7 | 2.7 |

5. 3.1 section: classification of study soils according to World Reference Base for Soil Resources could be useful for potential readers.

**Response:**

Thank you for your suggestion—we suggest adding a list of the most common soil types in the study area to the revised manuscript, as follow:

'Fluvisols, Leptosols, Lixisols, Luvisols and Vertisols are the prominent soil types in the study area (Dewitte et al., 2013)'

6. 3.1 section: in the PCA figure 2, it seems evident as DTPA-extractable concentrations are significantly and positively correlated with total concentrations of Zn in soil, while Ca(NO3)2-extractable concentrations are not. Can you try to explain this erratic behaviour?

**Response:**
DTPA extraction, a measurement of 'quantity', is expected to better correlate with total concentration in soil. On the contrary, the extraction in $Ca(NO_3)_2$, a measurement of 'intensity' or 'solubility', is more likely to depend on soil properties, chiefly soil pH. Therefore, it is not unexpected that Zn solubility (in 0.01 M Ca nitrate) is largely independent of the total concentration in soil whereas DTPA-extractable Zn reflects the total Zn concentration to a greater degree.

7. 3.4 section (line 341-343): consider adding this recent reference https://doi.org/ 10.3390/agronomy10091440

**Response:**
Thank you for your suggestion—we will add this reference to the revised manuscript.

8. Conclusions (line 402): use both/and. Alternatively, either/or.

**Response:**
Thank you—for more clarity, we will add 'or' to the revised manuscript at line 439.

---

## Author Comment (AC2)

**Responses to Referee 2 : MS No.: soil-2020-81**
Title: Zinc lability and solubility in soils of Ethiopia – an isotopic dilution study
Author(s): Abdul W. Mossa et al.

Many thanks for your time and effort in reading and commenting on our manuscript; your detailed points will help improve it. In the following we address the individual points you raise.

**Referee 2**

**General comments**

The paper presents interesting research on zinc lability and solubility in Ethiopia.

Understanding the behaviour of zinc in these soils is essential in combating zinc deficiency, which is an issue affecting a quarter of the population in Sub-Saharan Africa. The authors explain in detail the influence of soil properties on different measures of zinc lability and show that these properties are more important than the total zinc content. The conclusion is that soil acidity is by far the most dominant factor, and for some soils the organic matter content plays a role. This information is useful in designing soil management strategies to improve zinc availability. The manuscript does not have a clear multidisciplinary context compared to other articles in SOIL, nevertheless the results may be relevant for a broad international audience.

> **Response:**
>
> We agree with the referee that the manuscript has a fairly constrained focus on technical aspects of Zn geochemistry in soils within the SSA region. Nevertheless, as such, the manuscript has important agronomical, health and economic implications given the importance of Zn deficiency in human populations—especially within SSA.

My overall impression is that this paper is very well written and contains a lot of relevant information. I have no comments on the introduction. A major shortcoming was found in the methods, where too little attention is paid to developing an adequate geochemical model.

> **Response:**
>
> Description of the use of the geochemical model will be improved in the revised manuscript as described below in our responses to specific comments from the referee below.

Next, I feel the results and discussion section is generally very interesting but could benefit from better organization. My main concern here is that relatively little attention is paid to the explanation of the study's own results, as the paper regularly reads like a literature review without clear reference to what those citations mean for the interpretation of data.

> **Response:**
>
> An important, and relatively novel, part of our manuscript is the isotopic dilution method that we used. Consequently, the focus of the manuscript is partly on method development and validation. Therefore, we have presented our results alongside the weight of evidence on the subject from the literature.  This

is particularly important because our data for the SSA soils shows a distinct (and perhaps unexpected) trend in E-value with pH compared with previous studies.

In addition, the introduction clearly states the problem of zinc deficiency, and as such the implications for soil management should be part of the discussion, instead of only a few sentences in the conclusion. Overall, this is a high-quality paper that with moderate revisions would be suitable for publication.

**Response:**

This is addressed below in the specific comments.

**Specific comments**

The title suggests an isotopic dilution study, while ID is only part of the work. The importance of the study seems to be to explain the effect of soil properties, for which ID was part of the methods but not the sole or main method. It is recommended to include 'soil properties' in the title, and remove 'an isotopic dilution study'.

**Response:**

Thank you for your suggestion. We believe that a substantial part of the novelty of the manuscript lies in the measurement of isotopically exchangeable Zn in soils from the study area, and the development and validation of the isotopic dilution method in this application. Therefore, we think that this should be emphasised and reflected in the title. However, we suggest a modified title including 'the effect of soil properties', as suggested.

Line 72: what determined the time of oven drying between 24-48 hours, and did that have an influence on the results?

**Response:**

The time required was determined by the texture and moisture content of the soils but in practice, this was a judgement, based on experience rather than prescription. For greater clarity, we suggest adding the phrase 'depending on the moisture contents of the soil samples' to the revised manuscript.

Please write in full the abbreviations the first time they are used (e.g. VWP, NPOC).

**Response:**

Thank you—we will define the abbreviations in full in the revised manuscript.

Some more attention should be given to the modelling using WHAM7, especially with respect to the models for Al, Fe and Mn oxides. What type of model is used and on which specific oxides is the parameterisation of the models based? What are the assumptions with respect to the specific surface area.

**Response:**

We suggest adding the following to the revised manuscript:

> "The surface chemistry of oxides is simulated by a surface complexation model (Lofts and Tipping, 1998), which views the oxide surfaces as bearing hydroxyl groups that interact with protons and metal ions. Default values for specific surface areas in WHAM7 were used"

The input should be described more precise e.g. line 139 "Inputs to the model included cation and anion concentrations...." Specify which cations and anions and whether inputs are solution concentrations or concentrations in the soil solid phase.

**Response:**

Thank you for your suggestion—we suggest adding further details to the revised manuscript as follow:

> "Inputs to the model included solution concentrations of cations (Na, Mg, Al, K, Ca, Mn, Fe, Co, Ni, Cu, Zn, Cd, Ba, Pb, U) and the anions ($NO3^-$, $PO4^{-3}$) in the solution phase of the $Ca(NO_3)_2$ soil suspensions"

Do the cations include $Fe^{3+}$ and $Al^{3+}$? Tipping showed that these are important cations to consider because of their competition with other (trace) metals for binding to (dissolved) organic matter. In the case these cations were not measured their activities can be calculated from equilibrium with iron- and aluminium (hydr)oxide according to Tipping et al. 2002 (Geochimica et Cosmochimica Acta 66, 3211-3224)

**Response:**

Measured concentrations of $Fe^{3+}$ and $Al^{3+}$ in the solution phase of $Ca(NO_3)_2$ extraction were used as model input; this will be specified in the revised manuscript as described in response to your previous point.

A shortcoming of the study is the limited representativeness of the geochemical model for tropical (weathered) soils used to explain Zn lability and solubility. This stems from the assumptions made for the adsorptive constituents that are based on temperate soils, whereas the authors make it clear in the introduction that it is their ambition to study tropical soils, with the expectation that these will be different. This difference between tropical and temperate soils should then also be reflected in the model. The average fraction of humic substances was 36% for tropical soils modelled by Van Eynde et al. (2020): Boron speciation and extractability in temperate and tropical soils: A multi-surface modeling approach (Applied Geochemistry); this is notably different from the 50% used in the present study. In the study of van Eynde et al. it is also shown that the oxalate extractable Fe (non-crystalline oxides) is small compared to the dithionite extractable Fe (crystalline and non-crystalline) in such tropical soils. In the present study only oxalate extractable Fe is considered. Although the non-crystalline oxides have a much larger surface area than crystalline oxides, this may lead to an underestimation of the binding to iron oxides. Binding of metals to iron oxides is especially important at higher pH, which is the pH range for which the present study shows the highest overprediction of modelled soluble Zn.

**Response:**

The absence of characterization of soil organic matter (i.e. measurement of HA and FA) means that we are required to make certain assumption regarding the composition and reactivity of SOM. However, this is an approach which has been used frequently in previous studies and the 50% assumption that we make in the manuscript has been used extensively in the literature over a wide range of soils. The study you have cited may be qualitatively more relevant (HA 36% of SOM) but it reports the result for only 5 tropical

soils. Therefore, we suggest retaining our initial approach but we have cited the study and noted the uncertainty associated with making such assumption in the revised manuscript. As we, and others, have noted in previous studies, the principal weakness of applying mechanistic geochemical assemblage models to whole soils (probably) lies in the crude nature of the measurements of geocolloidal adsorbents (oxides, HA, FA etc) and the assumptions regarding their characteristics (surface area etc) rather than in the model itself which has been parameterised using single, well characterised geocolloidal constituents.

In response, we suggest adding the following to a revised manuscript:

> "Van Eynde et al., (2020) found that the average fraction of humic substances in 5 tropical soils was 36%. However, the composition and the reactivity of soil organic carbon will be different from one soil to another. Consequently, there will be an associated error when relying on such assumptions in the absence of full characterization of soil organic carbon"

A positive point is the modelling of binding of Zn to Mn oxides which is usually not considered in multi-surface modelling studies for soils (see review Groenenberg and Lofts, 2014 Environ. Toxicol. Chem. 33, 2181–2196). The study shows that Zn binding to Mn-oxides may be highly relevant according to model predictions.

**Response:**

Noted, thank you.

Line 170: indicate what could cause the discrepancy between observed values and those reported for contaminated and uncontaminated soils.

**Response:**

The differences between the results for E-values reported in this manuscript and those reported in the literature are discussed in detail in section 3.2.2

Line 172: indicate why it is relevant that concentrations were positively skewed

**Response:**

Positive skewness indicates that the concentration of DTPA extractable Zn is low in most studied samples. This is indicated in lines 196-197

Figure 1: the added value of this figure in addition to Table 1 is unclear. Only two references are made to it (Line 172 and Line 183), and the skewedness of the data is not explained to be that relevant that it deserves a full figure. Additionally, inferences can be made about the skewedness by comparing the median and mean values in the min-max range as is done in Table 1. It is suggested to remove the figure.

**Response:**

Thank you for your suggestion, but we believe that readers will appreciate the nature of the dataset from a graph that encapsulates the distributions of the four indices of Zn bioavailability in soil, in their entirety. Therefore, we suggest it is better to keep the current figure.

Figure 2: it is unclear which label belongs to which line. PCA was also not explained in the statistics. It is felt that if the objective is to 'evaluate the correlation between soil variables' (Line 186) a correlation matrix is more intuitive than a PCA graph.

**Response:**

Thank your suggestion. We will improve the labels in the graph in a revised manuscript. We think that PCA graphs provide a more comprehensive and dynamic representation of the data than a correlation matrix and would therefore like to retain the PCA graph. However, we suggest also adding a correlation matrix to the supplementary material, as suggested.

Line 209: please add a brief final sentence on the overall method assessment and validation step

**Response:**

We suggest adding the following summary statement to the revised manuscript:

'Thus, given (i) the robustness of the $^{70}$Zn distribution coefficient, (ii) the likelihood that the isotopic spike did not substantially affect the native Zn equilibrium or cause precipitation and (iii) the sub-micron filtration step and inter-laboratory agreement for Zn concentration in the 0.01 M Ca electrolyte soil suspensions, we are confident in the validity of the E-value determinations.

Line 220-223: unclear why this is relevant. In general, it is suggested to start with the most important explanations. It should be clearly explained why it is relevant to compare soils of the present study with urban or temperate agricultural soils when interpreting the pH effect.

**Response:**

Previous studies measuring isotopically exchangeable trace metals have been based on temperate soils (often contaminated) and urban soils. The weight of evidence from these studies suggests that %E-value (lability) declines with pH whereas the trend seen in our (uncontaminated) Ethiopian soils with their broad pH range differs from this. Therefore, it is important to discuss our results in the context of reported results in the literature and make comparisons that may potentially explain the difference between our results and those reported in previous studies.

Line 238-277: in the manuscript, this constitutes a one-page explanation of non-labile colloidal particles, leading to the conclusion that the correlation between ZnE and pH is genuine. This text can be shortened significantly. In addition, this part is more of a literature review, with relatively detailed accounts of the results found in other studies. What is missing is the link between the literature and the results found in the present study. For example, in line 266 the paragraph ends with the notion that solutions were filtered to 0.22 um; the authors fail to state what this means for their work and their data.

**Response:**

The study we cite (Tavakkoli et al., 2013) investigates the possible occurrence of non-isotopically exchangeable Zn in filtered soil suspensions. They found that the apparent proportion of isotopically exchangeable Zn was positively correlated with the pore size of the filter used (0.22, 0.45, 0.7 µm) - suggesting the presence of nano-particulate forms of Zn – whereas non-isotopically exchangeable Zn was absent when filtering to < 0.1 µm. As we used 0.22 µm filters in our study, we think that the nonisotopically exchangeable Zn in the filtered should be negligible. For greater clarity, the sentence "therefore, the existence of non-isotopically exchangeable Zn should be negligible." can be added to the revised manuscript.

Line 356-: In addition to the already mentioned possible explanations for overprediction also the presence of Zn-Al layered double hydroxides or Zn containing phyllosilicates could be considered (see already cited Bonten et al. 2008 and citations therein)

**Response:**

Thank you—we will cite this reference in the revised manuscript.

What is missing is a paragraph on the implications of the research for combating Zn deficiency. In the conclusion some 'tools' are mentioned, but this should be elaborated on at the end of the discussion. For example, it is concluded that it is soil properties rather than variation in total zinc that determines variability, but this is not translated in an overall conclusion on the conditions under which Zn deficiency occurs. A low solubility can still mean more Zn uptake if the total pool is larger, and vice versa.

**Response:**

This is a fair point. While our study does not specifically investigate crop uptake of Zn, we can address this in the Conclusions by re-working the text and adding an additional paragraph focussed on future work to link measurement of soil Zn availability with crop uptake. We suggest adding the following paragraph to the revised manuscript:

'Zinc uptake by crop plants will be determined by a combination of (i) soil Zn availability and (ii) plant controls over uptake and transport. It is likely that soil Zn availability will reflect both solubility and the magnitude of the reactive Zn reservoir that supports $Zn^{2+}$ ion activity in the soil solution. For the Amhara soils studied it appears that (i) DTPA-extractable Zn provides the best estimate of the available Zn pool controlling $Zn^{2+}$ solubility and (ii) that pH is the best predictor of the strength of labile Zn adsorption. Both assays are well recognised and fall within the scope of most agronomic laboratories. It therefore seems reasonable to suggest that $Zn_{DTPA}$ and soil pH may, in combination, be tested as predictor variables for Zn uptake by staple crops in considering both crop Zn deficiency and biofortification for human health'

**Technical corrections**

Line 14: either explain what 'major knowledge gaps' are meant, or more generally indicate that SSA soil types are understudied.

**Response:**

Thank you for your suggestion. This can be changed to "many soil types in SSA are poorly studied" in the revised manuscript.

Line 76: comma or and after 'was determined'.

**Response:** Thank you—we will add a comma to the revised manuscript.

Line 135: remove comma after 'any deviation'.

**Response:**

Thank you—we will delete the comma and reorganise the sentence for improved clarity.

Line 176: significant but weak positive correlation

**Response:**

Thank you for your suggestion—we will add the phrase 'but weak' to the revised manuscript.

Line 190: explain what is meant with 'react in opposite ways.

**Response:**

For greater clarity, we suggest replacing the phrase with: "PCA analysis also shows that $Zn_{Soln}$ and $Zn_E$ appear to be inversely correlated, which is mainly a consequence of their opposite trends with pH" in the revised manuscript.

Line 206: 'but' = and

**Response:**

Thank you— we will replace 'but' with 'and' in the revised manuscript.

Line 224: 'changes' = differences

**Response:**

Thank you— we suggest replacing 'changes' by 'differences' in the revised manuscript.

Line 235: start a new paragraph on non-labile particulate matter.

**Response:**

Thank you for your suggestion, we suggest a new paragraph is started in the revised manuscript.

Line 273: unclear what is meant with 'magnitude of the trend'.

**Response:**

Thank you—for greater clarity we suggest the phrase is changed to 'strength of the trend'.

Line 282-283: unclear why this is important for the present study, should be explained.

**Response:**

It is vitally important to point out that results of the fractionation simulated by WHAM are for the reactive/labile fraction of Zn estimated by isotopic dilution method. This distribution does not cover the (pseudo-) total concentration as measured by aqua regia digestion; non-labile or 'fixed' metal in soil is not included in the adsorption equilibria described by geochemical models such as WHAM.

Line 302: the authors should make it clear that the pH-trend for Kdlab contrasts the one found for ZnE, instead of leaving it for the reader to infer.

**Response:**

Thank you for your suggestion. The trend of $Zn_E$ with pH is discussed elsewhere (3.2.2). Although, Kd is one of two components in calculating $Zn_E$, they are different/separate. Therefore, we think there will be no added value to make such a contrast.

Line 316: the 'p' looks like rho.

**Response:**

Thank you for spotting that, this will be corrected to 'p' in the revised manuscript.

Line 316: 'some influence on metal adsorption strength would be expected', this is vague and should be explained more clearly. In general, the remaining sentences of this paragraph should be elaborated on, to further clarify the pH-effect, and contain a brief conclusion.

**Response:**

We simply mean that an effect of soil organic matter on metal solubility would be expected because of its importance as a metal adsorbent. However, the nature of the effect is complicated by the opposing effects of organic matter in the solid and solution phases – as discussed. In the revised manuscript we suggest changing the text to:

"some influence on metal adsorption strength would be expected because of the importance of humus as a metal adsorbent (Fan et al., 2016)."

The rest of the paragraph then explains the two opposing effects of organic matter in the solid and solution phases, which may account for the relatively weak overall effect on metal solubility.

Line 323: comma after 'strength of adsorption'

**Response:**

Thank you—we will add a comma to the revised manuscript

Line 361-365: This can be shortened, as all cited studies conclude the same.

**Response:**

Thank you for your suggestion. Even though the two references are similar we think they provide a slightly different explanation (e.g., specifying calcite and hydroxyapatite as binding phases). Therefore, we suggest it is better to keep them separate.

---

## Author Response (AR2)

**Second Responses to Referees: MS No.: soil-2020-81**

Title: Zinc lability and solubility in soils of Ethiopia – an isotopic dilution study Author(s): Abdul W. Mossa et al.

Thank you for your additional comments on our work.

**Referee 2**

The authors have addressed most comments and have made improvements of the original manuscript. As stated before, the manuscript contains much interesting data and findings. However, the major shortcoming in the development of geochemical models as expressed previously has, in my view, been insufficiently addressed.

The title, abstract and primary objectives do not mention a geochemical model. The focus thus far is on soil properties and isotopic dilution, and the main findings about Zn solubility (especially the pH effect) are based on correlation analysis.

**Response:** Thank you for your suggestion—we propose adding the following statements to the abstract and objectives respectively:

"A parametrised geochemical assemblage model (WHAM) was also employed to predict the solid phase fractionation of Zn in the soils under study, as an alternative to sequential chemical extractions."

"to investigate the best of three possible input parameters defining reactive Zn in tropical soils (isotopically exchangeable, DTPA-extractable, total) when using a parameterised geochemical assemblage model to predict Zn solubility—assumed likely to be the principal driver for plant uptake".

In the results, the last paragraph of 3.2.2 contains WHAM7 modelling, amongst others of the free Zn ion activity, which comes back in 3.5 ("Free Zn2+ activity in soil solution), but there the free Zn activity is shown to be pH-dependent based on simple correlation analysis. The section explicitly devoted to multi-surface modelling of soluble Zn (3.4) mainly evaluates the use of different inputs for geochemical models (ZnE, Zntot, ZnDTPA) in the prediction of Zn solubility. In contrast, in the conclusion the authors state "In this study combining (i) an isotopic dilution approach to determining reactive soil Zn (ZnE) and the relative strength of available Zn adsorption (Kd) with (ii) geochemical speciation modelling (WHAM7) provided useful insights into the intrinsic reactivity of Zn in SSA soils at a regional scale and revealed the key soil variables determining Zn solubility". In short, the role and importance of the geochemical model in the study is thus unclear.

Overall, the above the impression that geochemical models were used (i) to understand the role of soil properties and reactive surface in Zn solubility and (ii) in the comparison of different assays of Zn status. For both purposes, adequate model development and description is crucial.

**Response:** For better readability, we suggest reorganising the result and discussion section in the revised manuscript such that a subsection (3.4) is devoted solely to discussing geochemical modelling results.**

Particularly:

Compared to the original submission, a somewhat more detailed description of the geochemical model has been added. Nevertheless, it seems to remain a 'black-box' approach with reference to a model and not to a publication where the 'standard parameters' could be found. As far as I know. the publication on the oxide models in WHAM is Lofts and Tipping, An assemblage model for cation binding by natural particulate matter, Geochimica et Cosmochimica Acta, Vol. 62, No. 15, pp. 2609–2625, 1998 and has never been updated. There is little justification of choices made about the model (WHAM7) or about the selection of reactive surfaces.

**Response:** Several geochemical models which predict metal speciation in aqueous systems and soils have been developed. The intent behind using a geochemical model in the current work was not to compare different geochemical models nor to attempt to re-parameterise the WHAM model specifically. Rather, the purpose was simply to test three established measurements of Zn availability as alternative inputs to an established model (WHAM7), to predict Zn concentration in the soil solution, utilizing a large dataset from tropical soils.

The model WHAM7 was chosen because it has a long development heritage, it is commercially available in a fully parameterised form, has a proven track record and is widely used. For example, a simple search on the Web of Science database using the key words "WHAM" and "soil", between 2016 and 2020, yields 28 publications in which WHAM was used in a range of different scenarios. Thus, regarding the choice of reactive surfaces, we are obviously limited by the options that the model offers. However, we have acknowledged in the paper that there are potential reactive surfaces in calcareous soils that are not accounted for in WHAM which may help explain some trends in model fit observed.

• The authors rightfully acknowledge that the main weakness in geochemical models is the rough measurements of reactive surfaces that are used as input. I understand the choice of the authors to use the 50% humic substances assumption that 'has been used extensively in the literature over a wide range of soils'. Nevertheless, as a general comment, this wide range of soils is hugely underrepresented by tropical soils. As such, it is felt that it would have been a valuable addition to the paper to evaluate what would happen with estimates (e.g. of Zn solubility) if actually measured humic substances values are used (such as in Van Eynde). After all, to improve our understanding of tropical soils (an ambition explicitly stated by the authors in the introduction), we need to build on other studies.

**Response:** We agree with the referee on the importance of full characterization of adsorption surfaces, including those associated with soil organic matter and all forms of oxides (Fe, Al, Mn) and aluminosilicate clay minerals. Ultimately this may be necessary to fully validate input variables for geochemical models applied to all soils. Unfortunately, such an undertaking was beyond the scope of this work which utilized 465 spatially co-ordinated tropical soils. Nevertheless, to investigate the importance of error in quantifying organic adsorption surface capacity, we ran the model assuming 36% of the soil organic C to be 'active', as suggested by Van Eynde, rather than 50% as utilized in previous studies. The result of these simulations are presented below in Fig. S1. The *change* in  $Zn_{Soln}$  was highly pH dependent and ranged from -4.8% to -39.8% with a maximum disparity at soil pH 5.5. The trend shown in Fig. S1 (below) reflects the fractionation shown in Fig. 7B of the manuscript where HA dominates the fractionation of Zn in this pH range and so any change in estimated HA concentration has the greatest effect on Zn solubility. The effect of the composition of soil organic matter on model performance was assessed using the residual mean square error (RMSE) between measured pZnSoln and pZnSoln predicted by WHAM7 using different Zn bioavailability estimates as input parameters. The results of these comparisons are presented in Table 1.

Table 1. Residual mean squared errors of measured pZnSoln and predicted pZnSoln by WHAM7 assuming 50% of SOC active and 36% of SOM active.

| Input
variable | RMSE (50 % of SOC
assumed active) | RMSE (36 % of SOC
assumed active) |
|-------------------|--------------------------------------|--------------------------------------|
| Zn Tot | 2.63                                 | 2.70                                 |
| Zndtpa            | 0.42                                 | 0.48                                 |
| Znsoln            | 1.32                                 | 1.39                                 |

Figure S1. Relationship between soil pH and percent change in ZnSoln predicted by WHAM7 when assuming only 36%, rather than 50%, of soil organic C is active. ZnSoln was predicted by WHAM7 using isotopically exchangeable Zn, DTPA-extractable Zn, and total Zn concentration in soil as inputs.

We suggest adding Fig. S1 to the supplementary material and adding the following section to the revised manuscript:

"In common with previous studies, soil reactive organic matter was estimated as 50% of total soil organic C in all cases. However, in reality the composition and the reactivity of soil organic matter will differ between soils and there is currently uncertainty regarding the use of a universal value. Thus, for instance, Van Eynde et al., (2020) found that the average fraction of humic substances in 5 tropical soils was 36%. Consequently, there will be an associated error when relying on such assumptions in the absence of full characterization of soil organic matter in all soils. This is particularly important in the current study because while WHAM has been used and validated in several studies on temperate soils (Buekers et al., 2008; Izquierdo et al., 2013; Mao et al., 2017), to our knowledge, it has not been used on tropical soils. To test the effect of soil organic matter composition on model performance, we ran the model assuming 36% of soil organic C to be 'active' humic substances, as suggested by the data of Van Eynde et al., (2020). The results of these simulations are reported in Fig. S1 in the supplementary material. The change in Znsoln was pH dependent, ranged from –4.8% to -39.8% of the originally modelled value, and the maximum change occurred at soil c. pH 5.5. These results are relatively small changes considering that Znsoln covers over 4 orders of magnitude in the entire dataset".

My earlier comment on the importance of iron oxides (crystalline and non-crystalline) in metal binding at high pH (where overestimation of Zn solubility was found) was not addressed or incorporated in the discussion.

**Response:** Apologies for this omission. To test for potential effects of crystalline oxides on model performance, we ran the model using double the amount of Fe oxide estimated by oxalate extraction. This is probably an overestimate of Fe oxide adsorption capacity because the specific surface area of amorphous iron oxide may be 6x that of crystalline Fe oxide (Dijkstra et al., 2004). The ratio of crystalline to amorphous Fe oxide is clearly soil dependent but Van Eyndes et al., (2020) observed found that oxalate extractable Fe (amorphous) was smaller than dithionite extractable Fe (crystalline and amorphous) in tropical soils. Nevertheless doubling the Fe oxide content serves to test the sensitivity of the model to this variable. The results of these simulations are reported below in Fig. 2. The decrease in ZnSoln was up to 24% and was highly pH dependent where the effect was negligible at soil pH below 5.5. The model performance was assessed using the residual mean square error (RMSE) between measured pZnSoln and pZnSoln predicted by WHAM7 using different bioavailability Zn estimates as inputs to WHAM. The results of these comparisons are presented in Table 2. Table 2. Residual mean squared errors of measured pZnsoln and predicted pZnsoln by WHAM7 using measured amorphous Fe oxides and double the concentration of amorphous Fe oxides.

| Input
variable  | RMSE (measured
amorphous Fe oxides) | RMSE (double the concentration of
measured amorphous Fe oxide) |
|--------------------|----------------------------------------|-------------------------------------------------------------------|
| Zn Tot  | 2.63                                   | 2.60                                                              |
| Zn dtpa | 0.42                                   | 0.42                                                              |
| Zn Soln | 1.32                                   | 1.30                                                              |

We suggest adding Fig S2 (see below) to the supplementary material and adding the following section to the revised manuscript:

"Both amorphous and crystalline Fe oxide are involved in Zn binding in soil. The specific surface area of the crystalline Fe oxide has been assumed to be 1/6 of that of the amorphous Fe oxide (Dijkstra et al., 2004). However, it was previously found that oxalate extractable Fe (amorphous) was smaller than dithionite extractable Fe (crystalline and amorphous) in tropical soils (Van Eynde et al., 2020). For comparison, model performance was tested using double the amount of amorphous oxides as inputs. The results of these comparisons are presented in Fig S2 in the supplementary material. When doubling the amount of particulate Fe oxide in WHAM7 inputs, there was on average 3.42%, 5.43%, and 2.56% decrease in predicted  $Zn_{Soln}$  when using  $Zn_{E}$ ,  $Zn_{DTPA}$ , and  $Zn_{Tot}$  as inputs respectively. However, the change was highly pH-dependent, and although the effect was negligible at soil pH